# Overexpression *Populus* d-Type Cyclin Gene *PsnCYCD1;1* Influences Cell Division and Produces Curved Leaf in *Arabidopsis thaliana*

**DOI:** 10.3390/ijms22115837

**Published:** 2021-05-29

**Authors:** Tangchun Zheng, Lijuan Dai, Yi Liu, Shuang Li, Mi Zheng, Zhongnan Zhao, Guan-Zheng Qu

**Affiliations:** 1State Key Laboratory of Tree Genetics and Breeding, Northeast Forestry University, Harbin 150040, China; zhengtangchun@bjfu.edu.cn (T.Z.); dailijuan0060@163.com (L.D.); liuyi1992521@163.com (Y.L.); lishuang3726@163.com (S.L.); zhengmi0207@163.com (M.Z.); zhaozhongnanzzn@126.com (Z.Z.); 2National Engineering Research Center for Floriculture, School of Landscape Architecture, Beijing Forestry University, Beijing 100083, China

**Keywords:** *Populus*, *PsnCYCD1;1*, regulation of cyclin, small cell, curved leaf

## Abstract

d-type cyclins (CYCDs) are a special class of cyclins and play extremely important roles in plant growth and development. In the plant kingdom, most of the existing studies on CYCDs have been done on herbaceous plants, with few on perennial woody plants. Here, we identified a *Populus* d-type cyclin gene, *PsnCYCD1;1*, which is mainly transcribed in leaf buds and stems. The promoter of *PsnCYCD1;1* activated *GUS* gene expression and transgenic *Arabidopsis* lines were strongly GUS stained in whole seedlings and mature anthers. Moreover, subcellular localization analysis showed the fluorescence signal of PsnCYCD1;1-GFP fusion protein is present in the nucleus. Furthermore, overexpression of the *PsnCYCD1;1* gene in *Arabidopsis* can promote cell division and lead to small cell generation and cytokinin response, resulting in curved leaves and twisted inflorescence stems. Moreover, the transcriptional levels of endogenous genes, such as *ASs*, *KNATs*, *EXP10*, and *PHB*, were upregulated by *PsnCYCD1;1*. Together, our results indicated that *PsnCYCD1;1* participates in cell division by cytokinin response, providing new information on controlling plant architecture in woody plants.

## 1. Introduction

Flexible and precise control of cell division and proliferation is the key factor in organ evolution of higher plants. In order to understand the growth pattern of plants, it is necessary to reveal the relationship between cell cycle and growth regulation. The G1/S boundary is the key switch of the cell cycle process, which is functionally conserved between animals and plants and characterized as the d-type cyclin (CYCD)/retinoblastoma (Rb) pathway [1,2]. CYCDs are the rate-limiting components through the G1 phase [1]. In plants, the G1 phase of the cell division process was strongly affected by the expression level and activity of CYCDs under the coordinated regulation of sucrose and phytohormones [3,4,5].

Since three *Arabidopsis thaliana* CYCDs were first identified by complementation of a yeast G1 phase defective mutant strain [6], more CYCDs have been identified from various species [7,8,9]. However, most of them were characterized from herbaceous species like *Arabidopsis*, rice, and tobacco [10,11,12] Thus far, a total of ten *AtCYCDs* have been identified and divided into seven subfamilies (*CYCD1*-*CYCD7*) in *A. thaliana* [13]. Overexpression of *CYCD* genes promoted G1/S phase transition in *Arabidopsis* and there were strong interactions between most CYCDs and CDKA;1, verified by in vivo and in vitro experiments [14,15,16], indicating that the CDKA–CYCDs complex can regulate the cell division process. As for plant CYCDs, studies have been focused on genes in *Arabidopsis* and *CYCD2* and *CYCD3* have been researched the most. In fact, compared to wild type, the overexpression of the *AtCYCD2;1* gene in tobacco cannot change the size of each organ, but increases the growth rate of bud tips [17]. Furthermore, some *CYCD* genes have been verified with tissue-specific expression, two *CYCD3* genes in *Antirrhinum majus* showed different tissue expression specificity and *Am**CYCD3b* appears to be expressed in all dividing cells, whereas *Am**CYCD3a* is not expressed in stem tip meristem but is expressed primarily in the primordia that forms the lateral organ [18].

In perennial woody plants, 22 poplar *PtrCYCD* genes were identified by searching the *Populus trichocarpa* genome based on *Arabidopsis* CYCD proteins [19]. Woody plants have a much larger CYCD family than previously described in other species [20]. A rippled leaf poplar mutant was used for gene expression and microscopic analysis, and the *CYCLIN D1* gene was found to be responsible for this phenotype [21]. After being co-expressed in the vascular cambium of *Arabidopsis* roots, *Aintegumenta* (*ANT*) and *CYCD3;1* were essential for root secondary thickening in response to cytokinins [22]. *PtoCYCD3;3* transgenic poplar showed enlarged and wrinkled leaves and thickened and elongated stems, resulting in the abnormal differentiation of cambium and stem [23]. During the adventitious root development phase, *PeCYCD6;1* promoted the periclinal division and separated endodermis and cortex cell layers by association with the Short root (SHR)/Scarecrow (SCR) network [24].

In plants, most of the existing studies on *CYCDs* have been done on annual herbaceous plants, with few on perennial woody plants. In order to reveal the biological functions of *CYCDs* in woody plants, a novel *CYCD* gene (*PsnCYCD1;1*) of d-Type cyclins from *Populus simonii* × *P. nigra* was identified and performed a new function in transgenic *A. thaliana*.

## 2. Results

### 2.1. Isolation and Characteristic Analysis of PsnCYCD1;1

A d-type cyclin gene was successfully cloned by PCR using the total cDNA extracted from stems of double haploid poplar, which was named *PsnCYCD1;1* according to the homologous gene in *A. thaliana* and *P. trichocarpa*. As shown in Appendix A, *PsnCYCD1;1* of *P. simonii* × *P. nigra* and *PtrCYCD1;1* of *P. trichocarpa* had exactly the same length, which were both 987 bp with 20 base mutations at the nucleotide level and 6 aa substitutions at the amino acid level. PsnCYCD1;1 encoded a predicted protein of 328 aa with a molecular weight (MW) of 36.71 kDa and *pI* of 4.84.

Multi-sequence alignment analysis results showed that PsnCYCD1;1 shared the highest homology with *P. trichocarpa* (98%) and *Passiflora morifolia* (90%). The results of prediction on the Pfam and Expasy websites showed that these genes contained conserved regions of Cyclin-N/Cyclin-C domains and an LxCxE motif at the N-terminus, where Cyclin-box was the interaction region of the cyclin gene and cyclin-dependent kinase gene and the LxCxE motif functioned after being bound to the retinoblastoma gene (Appendix A).

### 2.2. Spatiotemporal Expression Patterns of PsnCYCD1;1 Gene

In order to investigate the expression pattern of *PsnCYCD1;1*, total RNAs were extracted from male and female flowers, leaf buds, shoots, young stems, differentiating xylems, young leaves, and roots. The results of qRT-PCR showed that the *PsnCYCD1;1* gene was expressed in eight different tissues, specifically, it was highly expressed in leaf buds, shoots, and stems, while exhibiting relatively high expression levels in differentiating xylem, young leaves, and roots (Figure 1A). For cross validation, the tissue-specific expression regions of *PsnCYCD1;1*, a 2.0 kb promoter fragment of *PsnCYCD1;1*, was cloned and constructed into a pBI121 vector with a *GUS* fusion gene and transferred into *A. thaliana* to generate transgenic seedlings. Finally, more than 15 transgenic lines harboring the *GUS* gene were obtained in total by PCR detection and GUS staining (Figure 1B–D). Promoter activity was strong in young seedlings, growing points, and young leaves. In the inflorescence, only the mature anthers were dyed blue. The above results indicated that the *PsnCYCD1;1* gene could be highly expressed in vigorous tissues, suggesting that the *PsnCYCD1;1* gene might be involved in cell division and growth.

### 2.3. Subcellular Localization of PsnCYCD1;1

Subcellular localization is crucial for the proper function of most proteins, and the constructed vector pUC-*PsnCYCD1;1*-*GFP* and control vector pUC-*GFP* were transiently transformed into onion epidermal cells by a gene gun. Confocal microscopy imaging showed that fluorescence signals of the control vector (GFP) were evenly distributed in the nuclei, cytoplasm, and cell membrane (Figure 2A–D), whereas fluorescence signals of PsnCYCD1;1-GFP could be observed only in the nucleus and overlapped with the DAPI signal, suggesting that PsnCYCD1;1 was a functional protein located in the nuclei (Figure 2E–G).

### 2.4. PsnCYCD1;1 Transgenic Arabidopsis Generate Curved Leaves and Twisted Inflorescence Stems

In order to investigate the functions of the *PsnCYCD1;1* gene in plants, the recombinant vector was transformed into *Arabidopsis* with a floral dip by an *Agrobacterium*-mediated method. Thirteen resistant seedlings were detected by PCR with genomic DNA extracted from young leaves (Appendix A). After germination on MS medium of T_2_ seeds for 8 days, there were significant differences between transgenic *Arabidopsis* and wild-type seedlings (Appendix A). The transgenic seedlings developed faster than the wild-type seedlings, the hypocotyl became longer (Appendix A), and the cotyledon and hypocotyl became curved. Moreover, positive lines were further confirmed by qRT-PCR (Figure 3A), which showed that the expression levels in different transgenic lines were different, which could be roughly divided into three types: (1) less than 10 times (L3, L4, L8), the transgenic lines could not be distinguish by the naked eye, which were tentatively named CK type; (2) more than 10 times and less than 50 times (L5, L6, L7, L10, L12), the transgenic lines had slightly curved leaves compared with wild-type seedlings, which were tentatively named Type I; (3) more than 50 times (L1, L2, L9, L11), the transgenic lines varied significantly from the wild-type seedlings, exhibiting completely curved leaves and twisted inflorescence stems, which were named Type II.

In order to observe the growth process of transgenic *Arabidopsis*, the seedlings were transplanted into the soil medium. The rosette leaves of wild-type plants were fully expanded, while the transgenic plants’ leaves were highly curled. Although the number of rosette leaves was not different from that of wild-type plants, the plant growth was weak (Figure 3B). After 1.5 months, the wild-type plants began to grow inflorescence meristems, while the transgenic plants were still in a vegetative growth state (Figure 3C). After continuous cultivation in long-day conditions for about 2 weeks, the inflorescences began to appear and the flowering time of each transgenic seedling was different. Moreover, the inflorescences were bent and they could not grow upright without artificial assistance (Figure 3D). The transgenic plants could also blossom and produce seeds normally, but the seed setting rate was very low.

In order to observe the difference of root growth and development between wild-type and transgenic *Arabidopsis*, 8-day-old seedlings were placed on solid MS medium and grew vertically. Two weeks later, the results showed that the roots of Type I and Type II transgenic *Arabidopsis* seedlings grew more slowly than those of the wild type (Appendix A). Although the lateral root development was not affected, the root elongation was very slow, and Type II was more severely affected than Type I. The root length of Type I was only 40% of that of the wild type, while that of Type II was only 25% of that of the wild type (Appendix A).

### 2.5. PsnCYCD1;1 Transgenic Arabidopsis Response to 6-BA

In order to investigate whether the transgenic *Arabidopsis* seedlings responded to plant hormones, eight-day-old wild-type and transgenic seedlings were transferred to medium containing 6-BA and NAA. The results showed that the hypocotyls of transgenic seedlings could be differentiated into callus on the medium containing only 6-BA and two hormones (Figure 4A,B), while the wild-type seedlings had no change in the same medium (Figure 4C,D).

In order to test whether having roots is a necessary condition for response to hormones, the roots of eight-day-old wild-type and the transgenic seedlings were cut off, and then seedlings were inserted into the MS medium containing different concentrations of auxin and cytokinin, with the medium without hormone as the control. After two weeks of treatment, wild-type and transgenic seedlings grew new roots at the base of the hypocotyl on medium without any hormone (Figure 4E). On the medium containing only 6-BA, there was a significant difference between transgenic and wild-type seedlings. The hypocotyl of transgenic seedlings differentiated into callus, while the wild-type seedlings did not differentiate (Figure 4F). On the medium only containing NAA, the roots of the transgenic seedlings were shorter and curved more than those of the wild type (Figure 4G). On the medium containing NAA and 6-BA, the phenotype of transgenic and wild-type seedlings was similar to that on the medium containing only 6-BA, but the wild-type seedlings grew a few roots (Figure 4H).

### 2.6. PsnCYCD1;1 Transgenic Arabidopsis Reduced Cell Size

In order to observe whether PsnCYCD1;1 affected the cell size of the transgenic seedlings, the rosette leaves of one-month-old seedlings were collected and the epidermal cells were torn off for microscopic observation (Figure 5). Compared with the wild type, the size of epidermal cells of Type II transgenic plants was less than half of that of the wild type, and the cell shape was more uniform (Figure 5C). However, there was no significant difference in the size of epidermal cells between Type I and wild-type plants (Figure 5A,B).

For further observe the microstructure of the cells on the surface of the seeds, the surface of the seeds was observed with a scanning electron microscope. The results showed there are some differences in the length of wild-type and transgenic seeds (Figure 5E–G). The statistics show that the Type I and Type II transgenic seeds are significantly longer than those of wild-type plants (Figure 5H). Electron microscopic observation showed that the cell density of transgenic seeds was higher than that of the wild type (Figure 5I–K). The statistical analysis showed that the surface area of Type I was not significantly different from that of wild-type plants, while the surface area of Type II was significantly different from that of wild-type plants (Figure 5L).

### 2.7. Induction Transcriptional Level of Morphogenesis of Leaf- and Stem-Related Genes

To detect the variations in transcription levels of cyclin-mediated downstream division genes and morphogenesis of leaf- and stem-related genes in transgenic *Arabidopsis*, total RNA of three-week-old seedlings from WT, Type I, and Type II plants were extracted and detected by qRT-PCR. Compared with the wild type, ten division-related genes were upregulated in all transgenic seedlings (Figure 6A). *AtE2F1*, *AtE2F2*, *AtE2F3*, *AtELP1*, *AtELP2*, *AtRb*, and *AtH4* genes were upregulated more than two times in Type I and Type II seedlings and, in particular, the highest expression of *AtELP1* was increased more than 10.5 times. However, *AtELP3*, *AtDPA*, and *AtDPB* genes were upregulated more than two times in Type II and 1.5 times in Type I.

Compared with the WT, *AtAS2* was upregulated more than two-fold in both Type I and Type II, and the highest expression level was up to five-fold. *AtKNAT1*, *AtKNAT2*, *AtAS1*, *AtANT*, *AtPHB*, and *AtPHV* were upregulated more than two times in Type II, but only 1.5 times in Type I. *AtEXP10* gene was upregulated more than two times in only one Type II transgenic line and 1.5–2.0 times in other samples. In addition, the expression level of *AtSTM* showed no significant difference in all the tested samples (Figure 6B).

## 3. Discussion

The cell cycle is the entire process from the completion of cell division to the completion of the next division, which can be subdivided into interphase (including G_1_ phase, S phase, and G_2_ phase) and mitotic phase. Flexible and precise control of cell division and proliferation is the key factor in organ evolution of higher plants. Thus far, a total of 32 cyclins have been discovered and can be divided into A, B, D, and H types, including 10, 11, 10, and 1 cyclin, respectively [1]. In the whole life cycle, each cyclin has its own spatiotemporal expression profile. In general, A-type cyclins (CYCAs) control the S-M phase process by itself, and then co-regulate the G_2_/M phase transformation and M phase process by forming complexes with B-type cyclins (CYCBs) [25,26,27]. G1/S transition of the cell division and cell proliferation processes were severely affected by the expression level and activity of CYCDs under the coordinated regulation of carbon sources and phytohormones [17,28]. In *A. thaliana*, 10 *AtCYCDs* were identified at the genome-wide level and can be divided into seven subtypes (*At**CYCD1*–*At**CYCD7*) [13]. In poplar, 26 *PtrCYCD* genes in *P. trichocarpa* were identified and divided into seven subtypes: five *PtrCYCD1* subgroups, two *PtrCYCD2* subgroups, six *PtrCYCD3* subgroups, four *PtrCYCD4* subgroups, three *PtrCYCD5* subgroups, five *PtrCYCD6* subgroups, and one *PtrCYCD7* subgroup [19]. In this study, the first D-type cyclin gene of *P. simonii* × *P. nigra*, was analyzed and named *PsnCYCD1;1*. According to homologous analysis at nucleic acid and amino acid levels, PsnCYCD1;1 and PtrCYCD1;1 are highly homologous (98%) with only a few amino acid differences (Appendix A). In higher plants and mammals, CYCDs generally contain an LxCxE domain at the N-terminus for activating G1/S transition of cell division by binding to Rb and phosphorylating Rb to release E2Fs [17,29]. In addition, CYCDs also contain a conserved Cyclin-box domain, which is used for forming a CDK-Cyclin complex with CDK [30]. Depending on the presence of PEST sequences, some CYCDs can be rapidly degraded through the ubiquitin protein degradation pathway [31]. In our study, PsnCYCD1;1 contained Cyclin-box, LxCxE, and PEST domains (Appendix A). Since the PEST sequence is involved in the proteolytic pathway, it is speculated that the presence or absence of PEST may affect the proteolytic activity. Protein localization can provide a reference for understanding the biological function of genes. PsnCYCD1;1 protein was located in the nucleus as a nucleoprotein, which is consistent with the localization of AtCYCD1;1, AtCYCD2;1, and AtCYCD3;1 in *Arabidopsis*, which are all localized in the nucleus [12,32].

After overexpression of the *PsnCYCD1;1* gene in *Arabidopsis*, transgenic plants were phenotypically differentiated. Specifically, the leaf edge curled inward and eventually formed a sack-like structure which completely covered the lower epidermis of the leaf (Figure 5A,B); the inflorescence twisted so that it could not stand upright (Figure 5C). These morphological phenotypes were somewhat similar to those of previous studies which indicated that the silencing of the *PtrHB7* gene in poplar resulted in leaf and stem curving [33]. Overexpression of the *CYCD3* gene in *Arabidopsis* increased the number of epidermal cells in transgenic plants and they formed curled leaves towards the medial axis. The results showed that transgenic plants had much smaller cells and a lower cell differentiation level compared with wild-type cells [34]. In the present study, the observation of leaf epidermal cells of transgenic leaves showed that transgenic lines had smaller and more cells than wild-type lines (Figure 5), suggesting that the *PsnCYCD1;1* gene accelerated the process of mitosis in transgenic plants. Overexpression of *AtCYCD2;**1* also reduced the size of epidermal cells and inhibited the endocycle in transgenic seedlings [35], which serves as evidence to prove that cell enlargement is a necessary condition for endocycle entry. Constitutively expressed *CYCD3 in Arabidopsis* promoted callus growth and propagation from leaf explants on MS medium without cytokinins [36], however, under normal conditions, both auxins and cytokinins are required. This phenomenon is consistent with the findings of Houssa et al., who reported that the application of cytokinins led to the activation of potential DNA replication origins, suggesting that CYCD3 is involved in the activation of DNA replication [37]. In our study, transgenic lines were treated with auxins and cytokinins. The results showed that there were significant differences between transgenic seedlings and wild-type seedlings on MS medium just containing 6-BA. The hypocotyl of transgenic seedlings was expanded into a callus, whereas the base of wild-type seedlings was not induced substantially. Moreover, there was a slightly significant difference between transgenic lines and wild-type lines on medium containing NAA, suggesting that the *PsnCYCD1;1* gene might also be involved in the initiation of cytokinin-activated DNA replication.

Plant growth and development may be controlled by the root apical meristem (RAM) and shoot apical meristem (SAM). The most active cell division occurs in the meristem. The E2F transcription factor is involved in the regulation of G_1_/S transition in animals and plants and is activated by regulating the cell cycle and transcription of many genes required for DNA replication [27]. E2F and dimerization partner (DP) protein form heterodimers that bind to the promoter of target genes of E2F [38,39], which are regulated by the negative regulator Rb [40]. Rb proteins could also bind to CYCDs and are phosphorylated by CYCD–CDKA complexes [41,42]. Mutants with knockout or downregulation of the *Rb* gene may exhibit excessive cell proliferation in the development of female gametes, root tips, and shoot tips and defects in cell differentiation [43,44]. Histone H4 is only expressed in S phase cells and is highly expressed in cells at the exponential growth phase [5]. The quantitative results showed that ten cyclin-mediated downstream division-related genes were upregulated in all the transgenic plants examined (Figure 6). These results indicate that the regulation of the plant cell cycle requires mutual cooperation of a large number of cyclin kinases, cyclins, and kinase inhibitors in transcriptional regulation, protein degradation, and the E2F–Rb pathway to maintain the balance of cell division and differentiation.

To investigate the reason for inflorescence curving in transgenic *Arabidopsis*, genes related to inflorescence growth were identified, including *EXP10*, *STM*, *KNAT*, *AS1*, *ANT*, *PHB*, and *PHV3*. Specifically, *EXP10* is a protein that promotes cell wall extension in plants [45]. There are two other *KNOX* genes in *Arabidopsis* (*KNAT1* and *KNAT2*), which are highly expressed in SAM but cannot be expressed in leaf primordium [46]. The *ASYMMETIC LEAVES2* (*AS2*) gene plays an important role in leaf development. It forms a protein complex with the protein product of AS1, which participates in the establishment of leaf polarity and maintains the closure of the *KNOX* gene in leaves [47]. The role of *STM* in maintaining SAM development is inhibited by the *AS1* gene. An *AS1* function-deficient mutant (*as1*) can form deeply cleaved leaves and sometimes develop stems on the leaves. In contrast, wild-type embryos can only express *AS1 in the cotyledon primordia, but stm* mutant embryos can express *AS1* at the predicted SAM site, indicating that *STM* can inhibit *AS1* expression [48]. In summary, due to antagonistic relationship between *STM* and *AS1*, SAM and leaf development processes proceed separately. The mutation of *phb* and *phv* can lead to the transformation from abaxial leaf to adaxial leaf, which plays a key role in the establishment of leaf polarity [49]. In this study, the expression levels of *KNAT1*, *KNAT2*, *AS1*, and *AS2* were significantly increased in transgenic *Arabidopsis*, while the *STM* gene was inhibited by ectopic expression of *AS* genes. Meanwhile, the expression levels of *AtPHB* and *AtPHV* were significantly upregulated, suggesting that overexpression of the *PsnCYCD1;1* gene promoted cell division and affected the development of leaf polarity.

In this study, the heterologous expression in *A. thaliana* may not always reflect the actual biology in native species. The main task in the future is to obtain a transgenic poplar to verify the phenotypic changes in leaves and stems. Furthermore, the regulatory network of the *PsnCYCD1;1* gene will be found by combining chip-seq technology with genomic prediction of expression profiles between transgenic and wild-type poplar.

In conclusion, this study characterized a *Populus* D-type cyclin gene, *PsnCYCD1;1*. Overexpression of the *PsnCYCD1;1* gene in *Arabidopsis* can result in curved leaves and inflorescence stem morphological changes and affect the transcriptional level of endogenous genes, providing new information on the control of plant architecture in woody plants.

## 4. Materials and Methods

### 4.1. Plant Material and Growth Conditions

Poplar tissue culture seedlings used for gene cloning were double haploid plants induced by pollen from a twenty-year-old *P. simonii* × *P. nigra* tree in Harbin, China. Different poplar tissues (flowers, leaf buds, stems, leaves, and roots) were collected for subsequent RNA extraction. Wild-type *A. thaliana* (Col-0) were grown in pots containing a mixture of turf peat, vermiculite, and pearlite (3:1:1 *v*/*v*) in an artificial climate chamber with an average temperature of 22 ± 2 °C, 16 h of light and 8 h of darkness, 200 µmol m^−2^ s^−1^ photons, and 60–75% relative humidity.

### 4.2. Isolation of Populus PsnCYCD1;1 and Sequence Analysis

Based on the *PtrCYCD1;1* gene sequence of *P. trichocarpa* from the NCBI database, the coding sequence of *PsnCYCD1;1* was cloned from a cDNA library constructed from young stems of *P. simonii* × *P. nigra* double haploid seedlings using RT-PCR with *ExTaq*^HS^ polymerase (TaKaRa, Dalian, China). The primers used to amplify *PsnCYCD1;1* are listed in Appendix A. Using a bioinformatics analysis platform based on Expasy tools (http://www.expasy.org/tools, accessed on 28 May 2021) to speculate on the function of the deduced PsnCYCD1;1 protein. Sequence alignments and the creation of a neighbor-joining (NJ) phylogenetic tree were performed using Bioedit (v7.0.4.1) [50] and MEGA7 (v7.0) [51] with 1000 bootstrap replicates.

### 4.3. Gene Constructs and Genetic Transformation

A 2 kb 5′-upstream promoter region of *PsnCYCD1;1* was cloned from *P. simonii* × *P. nigra* with *ExTaq*^HS^ polymerase (TaKaRa) according to the manufacturer’s instructions. After confirmation by sequencing, the promoter sequence was inserted into vector pBI121 to replace the *CaMV 35S* promoter sequence to obtain a GUS binary vector.

To express the *PsnCYCD1;1* gene in plants, the coding sequence of *PsnCYCD1;1* was inserted into the *Xba* I and *Sac* I sites of the binary vector pROKII, which contains a *CaMV 35S* promoter, *NOS* terminator, and a kanamycin resistance gene. Primers used for the construction of the plant expression vector are shown in Appendix A.

The resulting plant binary vectors were transferred into *Agrobacterium tumefaciens* strain GV3101 by the freeze-thaw transformation method [52] and transformed into *A. thaliana* with the floral-dip method [53].

### 4.4. Subcellular Localization Analysis of PsnCYCD1;1

The full-length *PsnCYCD1;1* coding region (without stop codon) was amplified from pMD18-T-*PsnCYCD1;1* by PCR, digested with *Sal* I and *Nco* I, collected the target fragments, and directionally ligated into vector pUC18 to construct the *CaMV 35S**::PsnCYCD1;1-GFP* fusion gene. The primers used to amplify *PsnCYCD1;1* are listed in Appendix A. The *PsnCYCD1;1-GFP* recombinants and empty vectors were transiently transformed into onion epidermal cells by the gene gun method (Bio-Rad PDS-1000/He System, USA). The green fluorescent signal of the *PsnCYCD1;1*-GFP fusion proteins was observed and photographed using laser confocal microscopy (model LSM410, Zeiss, Jena, Germany).

### 4.5. GUS Staining and Analysis

*Arabidopsis* seedlings were collected and incubated in GUS staining solution after proper vacuum filtration, and the dye solution consisted of 0.5 mM ferricyanide, 0.5 mM ferrocyanide, 10 mM EDTA, 20% methanol, 0.1% tritonX-100, 2 mM X-Gluc, and 100 mM sodium phosphate (pH 7.0). The GUS staining analysis process was as described by Jefferson et al. [54].

### 4.6. Quantitative Real-Time PCR Analysis

To detect the spatiotemporal expression pattern of *PsnCYCD1;1*, total RNA was isolated from different poplar tissues (including leaf buds, shoots, young stems, differentiating xylems, young leaves, roots) of a one-year-old plant and flowers of different genders from a plant more than ten years old with the MiniBEST Plant RNA Extraction Kit (TaKaRa). RNA quality and concentration were detected by gel electrophoresis and a spectrophotometer (Thermo Scientific, Wilmington, USA). The total RNAs were reverse transcribed into cDNA and qRT-PCR detection was performed with SYBR Premix *EX Taq* II (TaKaRa) on the MJ Opticon 2 System (Bio-Rad, Hercules, CA, USA). All the primers used for qRT-PCR are shown in Appendix A. The expression level in each sample was normalized using *Psn**actin* as a housekeeping gene and calculated based on the cycle threshold with the delta-delta CT method [55].

For transgenic plants, the total RNA was isolated from the young leaves (two weeks old) using the MiniBEST Plant RNA Extraction Kit (TaKaRa), then qRT-PCR analysis was performed with *Atactin* as a housekeeping gene. Each reaction was conducted in triplicate from independent plants to ensure reproducibility of results.

### 4.7. Histological and Morphological Microscope Observations

For morphological observation of leaf epidermal cells, two-week-old leaves from WT and transgenic *Arabidopsis* were collected and the epidermal cells were torn off with tweezers and put into water. The epidermal cells were placed on a slide with a drop of water, covered with a coverslip, and observed and photographed under a Zeiss light microscope (Docuval, Carl Zeiss, Germany).

To observe the cell morphology of seed coats, the seeds were placed on a shelving platform and mounted on a scanning electron microscope (FEI Quanta 200 FEG, The Netherlands) for observation. For TEM observation of stems, sections (50–60 nm) were cut and sprayed with a layer of gold powder. The scanning process was performed with the following operating parameters: high voltage 12.5KV; pressure >7.5e-3 Torr; filament current 2.34 A; emission current 97 μA.

### 4.8. Statistical Analyses

Statistical analyses were performed using SPSS 19.0 (SPSS, Chicago, Illinois, USA). ANOVA was used for multiple-group comparisons, and statistical significance (*p* < 0.05) was determined by a Student’s *t*-test.

## Figures and Tables

**Figure 1 ijms-22-05837-f001:**
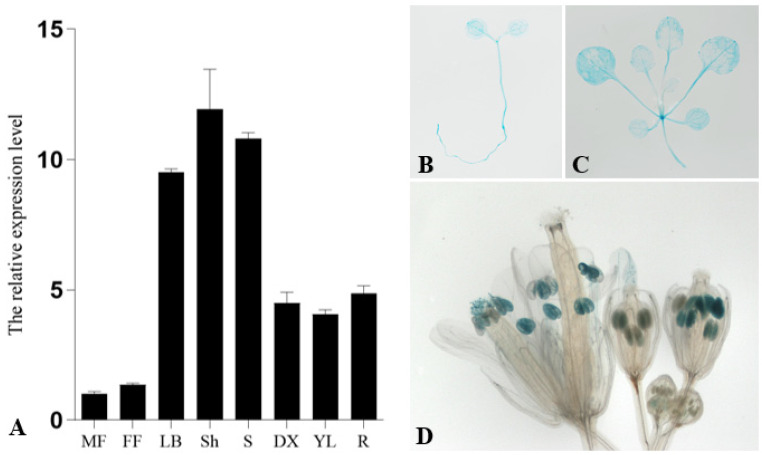
Spatiotemporal expression patterns of *PsnCYCD1;1* gene. (**A**) The relative expression level of *PsnCYCD1;1* gene in different tissues of *P. simonii* × *P. nigra*. MF: male flowers; FF: female flowers; LB: leaf buds; Sh: shoots; S: stems; DX: differentiating xylems; YL: young leaves; R: roots. Histochemical analysis of the GUS activity during different developmental stages of the *ProPsnCYCD1;1*::GUS transgenic *Arabidopsis*. GUS staining was observed in the 7-day old seedlings (**B**), three-weeks old seedlings (**C**), and the mature anthers in flowers (**D**).

**Figure 2 ijms-22-05837-f002:**
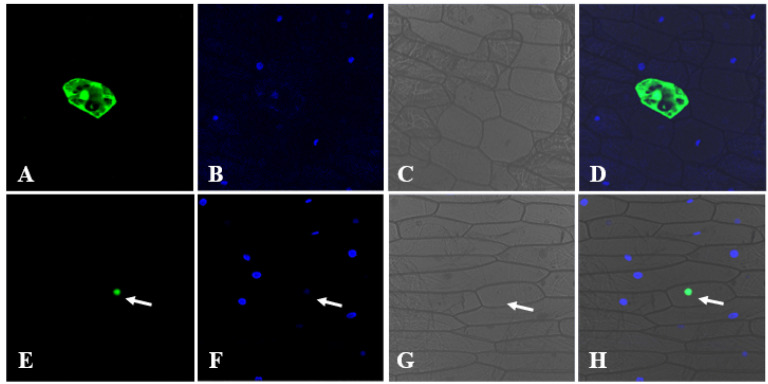
Subcellular localization of PsnCYCD1;1. The *35S::**PsnCYCD1;1-GFP* fusion vectors were transiently expressed in onion epidermal cells with *35S::GFP* vectors as control. The photographs were taken in dark field for green fluorescence (**A**,**E**), with DAPI signal (**B**,**F**), in bright light to examine cellular morphology (**C**,**G**), and in combination (**D**,**H**). The cells transiently expressed the GFP control (**A**–**D**) and PsnCYCD1;1-GFP fusion protein (**E**–**H**).

**Figure 3 ijms-22-05837-f003:**
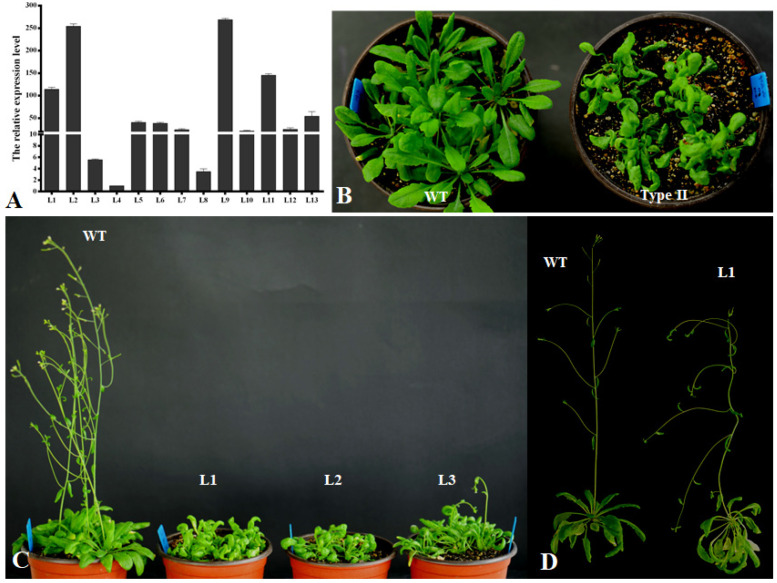
Curved leaf morphological feature of *PsnCYCD1;1* transgenic *Arabidopsis* seedlings. (**A**) qRT-PCR detection of transgenic *Arabidopsis*. L1-13, thirteen different transgenic lines. (**B**) One-month-old transgenic seedlings under short-day condition. (**C**) Two-month-old transgenic seedlings after transferring to long-day condition. (**D**) Sixty-day-old flowering wild type *Arabidopsis* and 95-day-old flowering transgenic plants. WT, wild type; Type II, Type II transgenic seedlings; L1-3, three independent transgenic lines.

**Figure 4 ijms-22-05837-f004:**
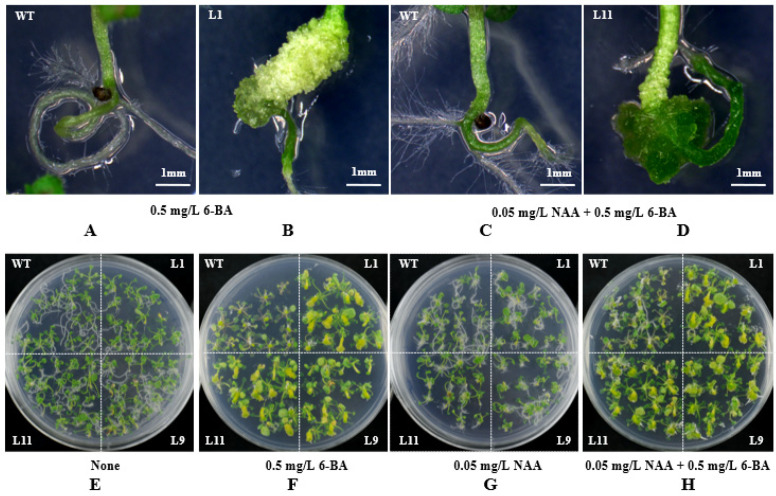
Response of transgenic seedlings to different hormones. (**A**,**B**) Enlarged view of wild-type and transgenic lines on MS medium containing 0.5 mg/L 6-BA. (**C**,**D**) Enlarged view of wild-type and transgenic lines on MS medium containing 0.05 mg/L NAA and 0.5 mg/L 6-BA. (**E**) MS medium without hormones. (**F**) MS medium containing 0.5 mg/L 6-BA. (**G**) MS medium containing 0.05 mg/L NAA. (**H**) MS medium containing 0.05 mg/L NAA and 0.5 mg/L 6-BA. WT, wild-type *Arabidopsis* seedlings; L1/9/11, three individual transgenic seedlings.

**Figure 5 ijms-22-05837-f005:**
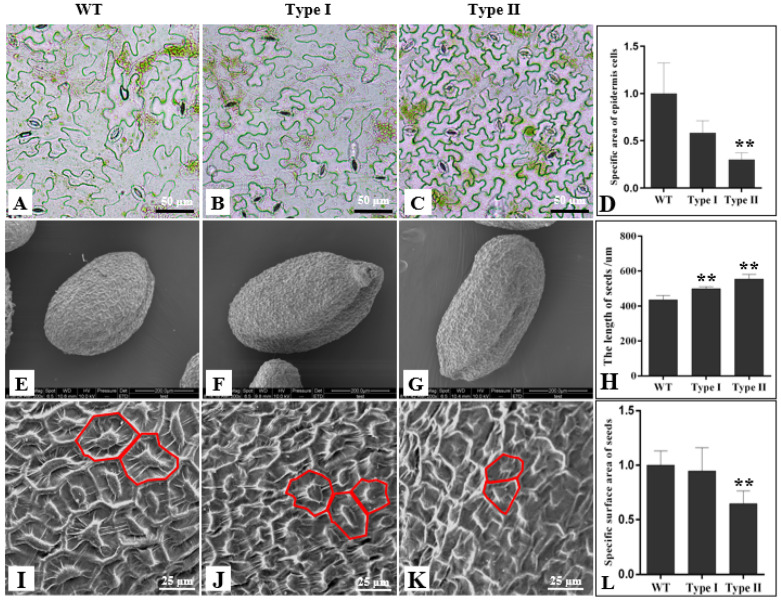
Observation of epidermal cells in transgenic *Arabidopsis.* Epidermal cells of wild type (**A**), Type I (**B**), and Type II (**C**) leaves. (**D**). Statistical analysis of the relative epidermal cell size between wild-type, Type I, and Type II leaves. SEM observation of cell size of wild type (**E**), Type I (**F**), and Type II (**G**) seeds. (**H**). Statistical analysis of the length of wild-type, Type I, and Type II seeds. SEM observation of epidermal cell size of wild-type (**I**), Type I (**J**), and Type II (**K**) seeds. (**L**). Statistical analysis of the epidermal surface area between wild-type, Type I, and Type II seeds. WT, wild-type seeds; Type I, Type I transgenic seeds; Type II, Type II transgenic seeds. ** *P* ≤ 0.01 with *t*-test.

**Figure 6 ijms-22-05837-f006:**
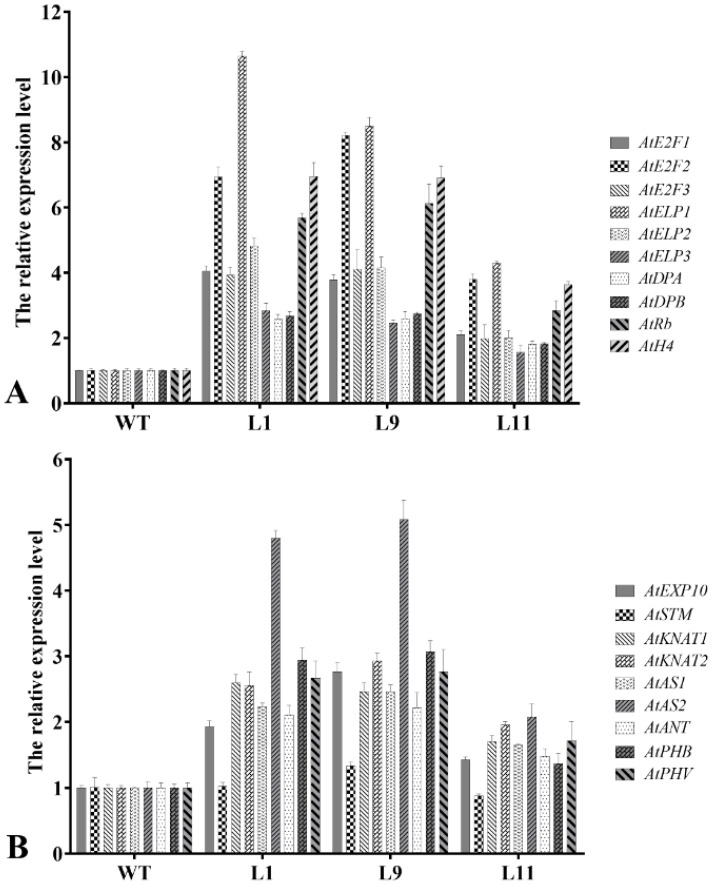
Transcriptional level of morphogenesis of leaf and stem-related genes. (**A**) Expression analysis of division-related genes in wild-type and transgenic *Arabidopsis* lines. (**B**) Expression analysis of curled leaf/stem-related genes in wild-type and transgenic *Arabidopsis* lines. WT, wild type; L1-3, three different transgenic lines.

## Data Availability

Not applicable.

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
