# Peer review of "Overexpression Populus d-Type Cyclin Gene PsnCYCD1;1 Influences Cell Division and Produces Curved Leaf in Arabidopsis thaliana"

_ijms, 2021, doi:10.3390/ijms22115837_

Round 1

Reviewer 1 Report

The manuscript by Zheng et al. provides a careful report on a comprehensive genetic expression test of Populus’ D-type cyclin gene PsnCYCD1-1 in Arabidopsis thaliana, and how its overexpression influences cell division and produces curved leaf. The work reads well, the experimental set up is pertinent, and the analytical steps are standard for this type of analyses. Main figures adequately summarize major results. Therefore, it definitively fits the standards of IJMS. However, before being able to recommend full acceptance, I request authors to address the following minor amendments. Title. Replace “;” by “-” 1. Introduction. At the end of the introduction section, and in order to clarify the null hypothesis/research gap, explicitly define research hypothesis, goals, and expected results. Refrain from describing the results. So far this last paragraph reads as an abstract and not as an introduction. 3a. Discussion. Include a brief discussion paragraph on how Populus’ D-type cyclin gene PsnCYCD1-1 interacts with other genes in the same pathway in additive and epistatic manners, which could lead to gene expression profiles’ linkage disequilibrium (refer to and mention “Tree Genet Genomes 2012 8(4):821-29” https://doi.org/10.1007/s11295-012-0467-x for an analogous case study in Populus referring the lignin pathway), as well as genetic correlations and trait’s trade-off via genes’ pleiotropy (amend accordingly by referring to and citing “Ecol Evol 2016 6(12):3940-52” https://doi.org/10.1002/ece3.2171 for a case study also within the Salicaceae family). 3b. Discussion. Add at the end of the discussion section, an explicit Conclusions’ subsection. Just before this new subsection, please also add a short paragraph discussing caveats of the present study, and needed steps (i.e. perspectives, also as a new subsection) for oncoming works. In the latter, authors should bridge the major gaps of the field in the light of the key achievements of the work. They must propose innovative ways to handle the caveats of the study, and suggest future key experiments and analyses (expression mapping via eGWAS, and genomic prediction of expression profiles). In this respect, refer to and cite modern works/reviews in tree species, including “Tree Genet Genomes 2021 17:12” (https://doi.org/10.1007/s11295-020-01489-1), and “Front Plant Sci 2020 11:583323” (https://doi.org/10.3389/fpls.2020.583323). 4. M&M. Concerning statistical analyses in subsection 4.8, an alternative model, more suited given the goals of the work, is to control for confounding effects within the framework of a mixed linear model. Since the current model is of course valid, I would not request authors to replace their datasets under a mixed model, but rather to include a short sentence justifying their decision and referring to supplemental material in which the MLM is described.

Author Response

Dear Reviewer,

Thank you very much for your letter and patient work for our manuscript. Those suggestions are all valuable and very helpful for revising and improving our paper. We have studied comments carefully and have made corrections that we hope to meet with approval. The manuscript was edited for proper English language, grammar, punctuation, spelling, and overall style by one of the English speaking editors at American Journal Experts. The main revises and the responses to comments are as follows:

The manuscript by Zheng et al. provides a careful report on a comprehensive genetic expression test of Populus’ D-type cyclin gene PsnCYCD1-1 in Arabidopsis thaliana, and how its overexpression influences cell division and produces curved leaf. The work reads well, the experimental set up is pertinent, and the analytical steps are standard for this type of analyses. Main figures adequately summarize major results. Therefore, it definitively fits the standards of IJMS. However, before being able to recommend full acceptance, I request authors to address the following minor amendments.

1. Title. Replace “;” by “-”.

Response: Thank you for your suggestions. In recent years, subfamily genes have been widely named by “;”. We prefer “;” for continuity of subsequent references. Such as ‘10.1105/tpc.110.080002’, “10.1016/j.pep.2019.105483”, “10.3390/ijms22031288”, “10.1007/s11295-015-0895-5”, “10.1242/bio.013128”.

  1.  Introduction. At the end of the introduction section, and in order to clarify the null hypothesis/research gap, explicitly define research hypothesis, goals, and expected results. Refrain from describing the results. So far this last paragraph reads as an abstract and not as an introduction.

Response: Thank you for your valuable suggestions. We had revised the last paragraph of the introduction section.

3a. Discussion. Include a brief discussion paragraph on how Populus’ D-type cyclin gene PsnCYCD1-1 interacts with other genes in the same pathway in additive and epistatic manners, which could lead to gene expression profiles’ linkage disequilibrium (refer to and mention “Tree Genet Genomes 2012 8(4):821-29” https://doi.org/10.1007/s11295-012-0467-x for an analogous case study in Populus referring the lignin pathway), as well as genetic correlations and trait’s trade-off via genes’ pleiotropy (amend accordingly by referring to and citing “Ecol Evol 2016 6(12):3940-52” https://doi.org/10.1002/ece3.2171 for a case study also within the Salicaceae family).

Response: Thank you for your valuable suggestions. We had revised the discussion paragraph and cited the references “Tree Genet Genomes 2012 8(4):821-29” and “Ecol Evol 2016 6(12):3940-52”.

3b. Discussion. Add at the end of the discussion section, an explicit Conclusions’ subsection. Just before this new subsection, please also add a short paragraph discussing caveats of the present study, and needed steps (i.e. perspectives, also as a new subsection) for oncoming works.

Response: Thank you. The “conclusion subsection” and the “perspectives subsection” were added in the end of discussion section.

  1. In the latter, authors should bridge the major gaps of the field in the light of the key achievements of the work. They must propose innovative ways to handle the caveats of the study, and suggest future key experiments and analyses (expression mapping via eGWAS, and genomic prediction of expression profiles). In this respect, refer to and cite modern works/reviews in tree species, including “Tree Genet Genomes 2021 17:12” (https://doi.org/10.1007/s11295-020-01489-1), and “Tree Genet Genomes 2021 17:12” (https://doi.org/10.3389/fpls.2020.583323).

Response: Thank you for your valuable suggestions. We had revised the discussion paragraph and cited the references “Tree Genet Genomes 2021 17:12” and “Frontiers in Plant Science, 2020, 11(583323)”.

  1. M&M. Concerning statistical analyses in subsection 4.8, an alternative model, more suited given the goals of the work, is to control for confounding effects within the framework of a mixed linear model. Since the current model is of course valid, I would not request authors to replace their datasets under a mixed model, but rather to include a short sentence justifying their decision and referring to supplemental material in which the MLM is described.

Response: Thank you for your valuable suggestions. We had revised the “M&M 4.8 subsection”.

Reviewer 2 Report

Zheng et al. present a study on the effects of overexpression of Populus D-type cyclin in Arabidopsis. They show that strong expression of this gene in Arabidopsis causes several effects including smaller cell size, curled leaves and induction of genes involved in cell division. Together, these results suggest that PsnCYCD1;1 is involved in cell division possibly through its effect on cytokinin pathways.

I think this was a sound study overall with the main caveat being that results from heterologous (and overexpression studies in general) expression experiments may not always reflect the actual biology in the native species. However, these results should be helpful for guiding future work on this gene.

Author Response

Dear Reviewer,

Thank you very much for your letter and patient work for our manuscript. Those suggestions are all valuable and very helpful for revising and improving our paper. We have studied comments carefully and have made corrections that we hope to meet with approval. The manuscript was edited for proper English language, grammar, punctuation, spelling, and overall style by one of the English speaking editors at American Journal Experts. The main revises and the responses to comments are as follows:

Zheng et al. present a study on the effects of overexpression of Populus D-type cyclin in Arabidopsis. They show that strong expression of this gene in Arabidopsis causes several effects including smaller cell size, curled leaves and induction of genes involved in cell division. Together, these results suggest that PsnCYCD1;1 is involved in cell division possibly through its effect on cytokinin pathways.

I think this was a sound study overall with the main caveat being that results from heterologous (and overexpression studies in general) expression experiments may not always reflect the actual biology in the native species. However, these results should be helpful for guiding future work on this gene.

Response: Thank you for your valuable suggestions. The shortage of this studies were analyzed and the future key experiments and analyses were discussed in the “Discussion section”.
